

# From Rindler fluid to dark fluid on the holographic cutoff surface

**Rong-Gen Cai[1,2], Gansukh Tumurtushaa[3,4] and Yun-Long Zhang[5,2,6⋆]**

**1** Institute of Theoretical Physics, Chinese Academy of Sciences(ITP-CAS) and School of Physical Sciences, University of Chinese Academy of Sciences, Beijing, China
**2** School of Fundamental Physics and Mathematical Sciences, Hangzhou Institute for Advanced Study, University of Chinese Academy of Sciences, Hangzhou, China
**3** Center for Quantum Spacetime(CQUeST), Sogang University, Seoul, Korea
**4** Center for Theoretical Physics of the Universe, Institute for Basic Science, Daejeon, Korea
**5** National Astronomy Observatories, Chinese Academy of Science, Beijing, China
**6** Center for Gravitational Physics, Yukawa Institute for Theoretical Physics(YITP), Kyoto University, Kyoto, Japan

⋆ zhangyunlong@nao.cas.cn

## Abstract

As an approximation to the near horizon regime of black holes, the Rindler fluid was proposed on an accelerating cutoff surface in the flat spacetime. The concept of the Rindler fluid was then generalized into a flat bulk with the cutoff surface of the induced de Sitter and FRW universe, such that an effective description of dark fluid in the accelerating universe can be investigated.

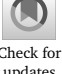

## 1 Introduction

The origin and properties of the dark fluid, mainly including the dark energy and dark matter, are still mysterious in the current universe. The model of Lambda Cold Dark Matter (ΛCDM) treats dark energy as the cosmological constant and dark matter as the collision-less particles, and explains the cosmic evolution and large-scale structures well. However, the tension between local measurements of the Hubble constant and the Planck's observation based on ΛCDM model becomes more important [1, 2]. Besides, the dark matter particles have not been detected directly. Thus, alternative models of the dark fluid such as modified gravity need to be reconsidered. One recent example is the emergent gravity by Verlinde [3], which is inspired by the volume law correction to the entropy on a holographic screen, whereas the Einstein gravity is related to the area law [4].

So is there a model which can unify these two scenarios of dark fluid and modified gravity? In this article, we show that a holographic model of the emergent dark universe (hEDU) can naturally realize the duality between the dark fluid in (3+1)-dimension and a modified gravity in (4+1)-dimension. We consider that the dark fluid in the universe emerges as the holographic stress-energy tensor on the hypersurface in one higher dimensional flat bulk [5, 6]. After adding the localized stress-energy tensor $T_{\mu\nu}$ on the hypersurface with intrinsic metric $g_{\mu\nu}$ and extrinsic curvature $\mathcal{K}_{\mu\nu}$, the induced Einstein field equations on the holographic screen are modified as

$$R_{\mu\nu} - \frac{1}{2}g_{\mu\nu}R = \kappa_4\big(T_{\mu\nu} + \langle\mathcal{T}\rangle_{\mu\nu}^d\big), \tag{1}$$

where $\langle\mathcal{T}\rangle_{\mu\nu}^d$ denotes the induced Brown-York stress-energy tensor [7],

$$\langle\mathcal{T}\rangle_{\mu\nu}^d \equiv \frac{1}{\kappa_4 L}\big(\mathcal{K}_{\mu\nu} - \mathcal{K}g_{\mu\nu}\big). \tag{2}$$

Here, $\kappa_4 = 8\pi G_4/c^4$ is the Einstein constant and the length scale $L = \kappa_5/\kappa_4$ is related to the positive cosmological constant $\Lambda = 3/L^2$. At the cosmological scale, we assume that $T_{\mu\nu}$ only includes the components of normal matter, and $\langle\mathcal{T}\rangle_{\mu\nu}^d$ represents the total dark components in our universe, such as dark energy and dark matter. The stress-energy tensor $\langle\mathcal{T}\rangle_{\mu\nu}^d$ as we formulated is similar to the Verlinde's elastic response of emergent gravity [3], in the way that it will back react on the background geometry.

The using of the Brown-York stress-energy tensor in (2) is inspired by the Wilsonian renormalization group (RG) flow approaches of fluid/gravity duality [8–14]. Where the holographic stress-energy tensor on the holographic cutoff surface is identified with the stress energy tensor of the dual fluid directly. When taking the near horizon limit, one can reach the so-called Rindler fluid [15–22], which is a new perspective on the membrane paradigm of black holes, where the Brown-York stress-energy tensor is used.

## 2 Dark Fluid on Holographic Cutoff

To see more clearly how the Einstein equation (1) works, it is interesting to consider a de Sitter hypersurface as the holographic screen in flat spacetime firstly. Then the dual stress tensor could contribute to the dark energy as $\langle\mathcal{T}\rangle_{\mu\nu}^\Lambda = -(\rho_c\tilde{\Omega}_\Lambda)g_{\mu\nu}$. After adding the baryonic matter with typical 4-velocity $u_\mu$ and stress-energy tensor $T_{\mu\nu} = (\rho_c\tilde{\Omega}_B)u_\mu u_\nu$ on the screen, both of dark matter and dark energy can be described by the stress-energy tensor of holographic dark fluid $\langle\mathcal{T}\rangle_{\mu\nu} = \langle\mathcal{T}\rangle_{\mu\nu} + \langle\mathcal{T}\rangle_{\mu\nu}^D$, where $\langle\mathcal{T}\rangle_{\mu\nu}^D = (\rho_c\tilde{\Omega}_D)\big[(1 + \tilde{w}_D)u_\mu u_\nu + \tilde{w}_D g_{\mu\nu}\big]$ and $\tilde{w}_D$ is the equation of state of the emergent dark matter. From the Hamiltonian constraint equation in higher dimensional spacetime, an interesting relation between these components can be derived [5],

$$\text{hEDU}: \quad \tilde{\Omega}_D^2 = \frac{\tilde{\Omega}_\Lambda}{2(1 + 3\tilde{w}_D)}\big[\tilde{\Omega}_D(1 - 3\tilde{w}_D) - \tilde{\Omega}_B\big]. \tag{3}$$

Once setting $\tilde{w}_D = 0$, we can compare (3) with the $\Lambda$CDM parameterization and it is straightforward to take the values from the observational data by Planck collaboration [23]. The toy constraint relation (3) can be satisfied within the margin of error $\Omega_D^2 - \frac{1}{2}\Omega_L(\Omega_D - \Omega_B) \lesssim 1\%$. After considering $1 \simeq \Omega_L + \Omega_B + \Omega_D$, we also have $\Omega_B \simeq \Omega_D - 3\Omega_D^2 - \Omega_B^2$. In order to see this relation more clearly we plot it in Fig. 1, together with Verlinde's relation $\Omega_B = \frac{3}{4}\Omega_D^2$.

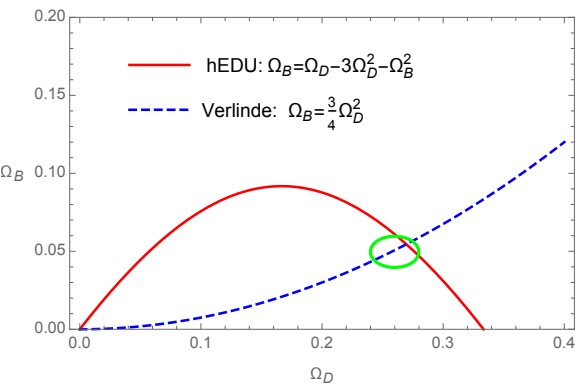

Figure 1: The schematic diagram of the relations between the components of baryonic matter $\Omega_B$ and dark matter $\Omega_D$ in the present universe. The green circle indicates the rough regime from the observation with $\Omega_B \simeq 0.05 \pm 0.01, \Omega_D \simeq 0.26 \pm 0.02$.

## 3   Modified Friedmann equation

The consistent embedding of a Friedmann–Lemaitre–Robertson–Walker (FLRW) universe in $4+1$ dimensional flat spacetime has been studied in [24,25]. In the spirit of the membrane paradigm [26,27], we remove half part of the bulk spacetime, which can be effectively replaced by the holographic stress tensor $\langle \mathcal{T} \rangle^d_{\mu\nu}$ in (2). The energy density and pressure in $\langle \mathcal{T} \rangle^d_{\mu\nu}$ are calculated to be $\rho_d(t) = \rho_c \sqrt{\Omega_L} \sqrt{\frac{H(t)^2}{H_0^2} + \frac{\Omega_I}{a(t)^4}}$, where the critical density and other parameters are given by $\rho_c = \frac{3H_0^2 M_P^2}{\hbar c}$, $\Omega_L = \frac{c^2}{L^2 H_0^2}$ and $\Omega_I \equiv \frac{Ic^2}{L^2 H_0^2}$. Considering the relation between the redshift $z$ and the scale factor via $a(t)/a(t_0) = 1/(1+z)$, we arrive at the normalized Hubble parameters $H(z)/H_0$ in terms of the redshift $z$, which is the modified Friedmann equation in the hEDU model,

$$\frac{H(z)^2}{H_0^2} = \frac{\Omega_L}{2} + \Omega_m(1+z)^3 + \Omega_r(1+z)^4 + \frac{\Omega_L}{2}\sqrt{1 + \frac{4}{\Omega_L}\Big[\Omega_m(1+z)^3 + (\Omega_r + \Omega_I)(1+z)^4\Big]}. \quad (4)$$

Notice here that at the current universe $z = 0$, we have $1 = \Omega_m + \Omega_r + \sqrt{\Omega_L(1+\Omega_I)}$, and we will consider the fact that the radition compoents $\Omega_r \ll 1$. By setting $\Omega_I = 0$, we can recover the usual Friedmann equation of the self-accelerating branch of the DGP braneworld model (sDGP) [28, 29]. When $\Omega_I \ll 1$, the behavior of $\Omega_I(1+z)^4$ is more like the dark radiation [30]. However, in this hEDU model, $\Omega_I \gg \Omega_r$ turns out not to be so small, such that the whole dark sector, including dark energy and apparent dark matter, is expected to be included in the holographic dark fluid [5]. In Fig. 2, we plot the equation of state parameter of the holographic dark fluid $\tilde{w}_d(z)$ in terms of the redshift $z$, as well as the $\tilde{w}_D(z)$ of apparent dark matter where the effective components of cosmological constant $\Lambda$ has been deducted.

In [6], the Markov-chain Monte Carlo (MCMC) sampling analysis together with the observational data of Type Ia supernovae (SNIa) and the direct measurement of Hubble constant $H_0$ [32] are employed. The two-dimensional observational contours are plotted in Fig. 3, with the 1-3$\sigma$ confidence contours for various parameters in the hEDU model [6]. The best-fit values turn out to be $\Omega_I = 0.43 \pm 0.13$ and $\Omega_m = 0.03 \pm 0.05$. The matter component is small enough and matches well with our theoretical assumption that only the normal matter is required.

We comment on the possible constraints from gravitational wave observations. It is argued that in general the modified gravity models are constrained from two aspects [33]. One is the constraint of the energy loss rate from ultra high energy cosmic rays, which indicates

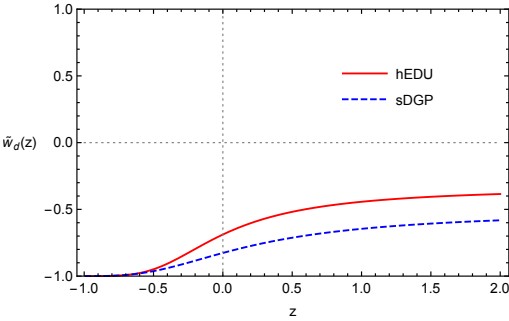 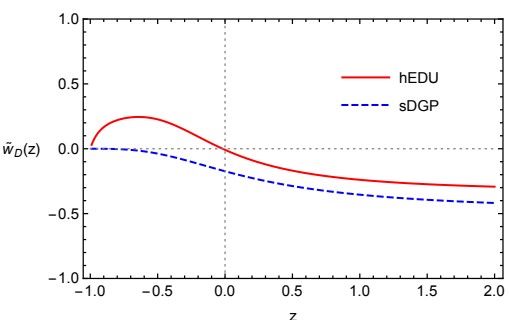

Figure 2: Left: the equation of state of the holographic dark fluid $\tilde{w}_d(z)$ in terms of the redshift $z$. Right: the equation of state of apparent dark matter $\tilde{w}_D(z)$, after deducting an effective cosmological constant. We adopt the following value for sDGP: $\Omega_I = 0$, $\Omega_m = 0.21$ [31] and hEDU: $\Omega_I = 0.4$, $\Omega_m = 0.04$ [6].

that gravitational waves should propagate at the speed of light. The other is the observed gravitational waveforms from LIGO, which are consistent with Einstein's gravity and suggest that the gravitational wave should satisfy linear equations of motion in the weak-field limit. For our model, the Bianchi identity leads to $0 \equiv \nabla^\mu G_{\mu\nu} = \kappa_4 \nabla^\mu T_{\mu\nu} + \kappa_4 \nabla^\mu \langle \mathcal{T} \rangle_{\mu\nu}$. If we do not put additional sources in the bulk, the Brown-York stress-energy tensor (2) itself is conserved $\nabla^\mu \langle \mathcal{T} \rangle_{\mu\nu} = 0$. Thus, it is similar to the effects of particle dark matter and it does not conflict with the observations from LIGO so far [34].

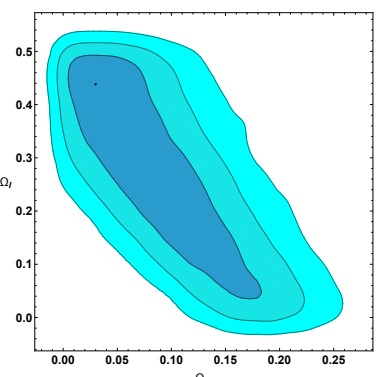 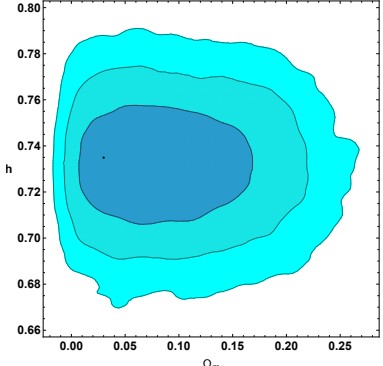

Figure 3: The 1-3$\sigma$ confidence contours for various parameters in the hEDU model, $\Omega_m$, $\Omega_I$, $h = H_0/(100\,\mathrm{km\,s^{-1}\,Mpc^{-1}})$, with figures taken from [6]. It is based on the MCMC sampling analysis with the observational data of Type Ia supernovae (SNIa) and the direct measurement of Hubble constant $H_0$.

## 4 Summary and Discussions

In summary, we construct a model of the dark fluid in our universe, which originates from the holographic stress-energy tensor $\langle \mathcal{T} \rangle^d_{\mu\nu}$ of higher dimensional spacetime. The toy hEDU model on a de-Sitter screen in flat bulk spacetime produces one additional constraint from $\Lambda$CDM parameterization to the components of the late-time universe. We derive the corresponding Friedmann equation and present a good fitting result with the observational data. Finally, we would like to mention the literature on modified Newtonian dynamics (MOND) from a brane-world picture [35, 36], as well as the holographic big bang model in [37, 38] which describes the early universe with a 3-brane out of a collapsing star in (4+1)- dimensional bulk. These

concepts are all related to our setups in the hEDU model. These models propose a possible origin of dark matter and dark energy and shed light on the underlying construction of the universe.

Finally, we discuss and comment more on our motivation and the details of this model. One may ask, is there an action in the (4+1) dimension from which equation (1) and (2) can be derived? The answer is yes! As has been shown in section 3 of [5], for the most simple case with a Minkowski bulk, the action is the same as that in the Dvali-Gabadadze-Porrati (DGP) brane world model [28],

$$ S_5 = \frac{1}{2\kappa_5} \left( \int d^5 x \sqrt{-g_5} R_5 + \int d^4 x \sqrt{-g_4} \mathcal{K} \right) + \frac{1}{2\kappa_4} \int d^4 x \left( \sqrt{-g_4} R_4 + 2\kappa_4 \mathcal{L}_M \right), \qquad (5) $$

which will lead to the equation of motion in (1). However, compare with the old braneworld scenario, here we have a different motivation. Our scenario is inspired by the fluid/gravity duality on the finite cutoff surface, which was proposed in [8, 9], that the Brown-York stress-energy tensor on the cutoff surface in Rindler spacetime $T_{\mu\nu}^{BY} = \frac{1}{\kappa_4 L} \left( \mathcal{K}_{\mu\nu} - \mathcal{K} g_{\mu\nu} \right)$ is identified as the stress-energy tensor of the dual fluid in lower dimension. More generalizations on the cutoff approaches to the fluid/gravity duality can be found in [10–14]. More interestingly, we have shown the duality between two viewpoints on the dark matter. From the 5-dimensional point of view, there is indeed an extra dimension, which belongs to the modified gravity. However, from the 4-dimensional point of view, the dark matter is described by the holographic stress-energy tensor, which is still a kind of matter, but only have the gravitational interaction with the standard model sector.

# Acknowledgements

We thank Sunly Khimphun, Bum-Hoon Lee and Sichun Sun for the collaboration on relevant topics. R. -G. Cai was supported by the National Natural Science Foundation of China (No.11690022, No.11435006, No.11647601,No. 11851302, and No. 11821505), Strategic Priority Research Program of CAS (No. XDB23030100), Key Research Program of Frontier Sciences of CAS; G. Tumurtushaa was supported by the Institute for Basic Science (IBS) under the project code(IBS-R018-D1) and CQUeST at Sogang University; Y. -L. Zhang was supported by Grant-in-Aid for JSPS international research fellow(18F18315).

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
