# Peer review of "From Rindler Fluid to Dark Fluid on the Holographic Cutoff Surface"

_SciPost Physics Proceedings, doi:SciPost Phys. Proc. 4, 003 (2021)_

## Round 1 · Referee Report · Anonymous (Referee 1) · 2020-11-15

Strengths
1- ambitious 2- attacking a very relevant problem 3- concise
Weaknesses
1- theoretical derivation quite speculative 2- idea not too innovative (cf Verlinde's work) 3- motivation of the framework disputable
Report
The paper attacks the fundamental problem of dark matter and dark energy following the original idea of Verlinde and building on it. The presentation is adequate. My major complaint is about the formal derivation and motivation for Eq(1)-(2) which constitute the basis of the framework.
Looking at equation (1) it seems that the idea is to embed our world (the 4D brane) in a 5D spacetime and assume that dark matter comes from the matter degrees of freedom in the extra dimension. Despite this seems quite an old idea (cf. Randall-Sundrum type models), its motivation and implementation is not so robust. Eg. Is there an action in 5D from which Eq.(1) can be derived?
Also, in principle, the stress tensor contribution coming from the extra dimension could be more complicated that what used in Eq(2) and it is strongly dependent on the ''coupling'' between our 4dimensional world and the fields living in the 5dimensional bulk. What does motivate such a simple form?
Is this related to the idea that DM comes simply from the gravitational dynamics and not from any hidden matter sector couple to the standard model?
In summary, I urge the authors to introduce better the model and the ideas behind the theoretical formulation. Once Eq(1)-(2) are accepted, the computations and the paper flow quite nicely, but those are key points whose assumptions and derivation have to be explained in more detail for the Reader not totally familiar with the topic.
Looking at equation (1) it seems that the idea is to embed our world (the 4D brane) in a 5D spacetime and assume that dark matter comes from the matter degrees of freedom in the extra dimension. Despite this seems quite an old idea (cf. Randall-Sundrum type models), its motivation and implementation is not so robust. Eg. Is there an action in 5D from which Eq.(1) can be derived?
Also, in principle, the stress tensor contribution coming from the extra dimension could be more complicated that what used in Eq(2) and it is strongly dependent on the ''coupling'' between our 4dimensional world and the fields living in the 5dimensional bulk. What does motivate such a simple form?
Is this related to the idea that DM comes simply from the gravitational dynamics and not from any hidden matter sector couple to the standard model?
In summary, I urge the authors to introduce better the model and the ideas behind the theoretical formulation. Once Eq(1)-(2) are accepted, the computations and the paper flow quite nicely, but those are key points whose assumptions and derivation have to be explained in more detail for the Reader not totally familiar with the topic.
Requested changes
1- justify and introduce better the theoretical background, the assumptions and the physical ideas behind equations 1 and 2.

---

## Round 2 · Referee Report · Anonymous (Referee 1) · 2020-12-2

Report

I would like to thank the authors for their reply.
I am happy with them.
I feel comfortable with suggesting the publication of the manuscript.

---

## Round 2 · Author Response

Thanks a lot for this very nice report! Please see the responses below, which are also added as the last paragraph of section 4, together with eq.(5).

---

## Round 2 · List of Changes

1). Is there an action in 5D from which Eq.(1) can be derived? Re 1). Yes! There is an action for this, which is similar to the DGP braneworld model, that has been added in eq.(5).

2) What does motivate such a simple form? Re 2) It is inspired by the fluid/gravity duality on the finite cutoff surface, in references [8-9].

3) Is this related to the idea that DM comes simply from the gravitational dynamics and not from any hidden matter sector couple to the standard model?

Re 3). The answer is a bit subtle but interesting. We have shown the duality between two viewpoints on the dark matter. From the 5 dimensional point of view, there is indeed an extra dimension, which belongs to the modified gravity. However, from the 4-dimensional point of view, the dark matter is described by the holographic stress-energy tensor, which is still a kind of matter, but only have the gravitational interaction with the standard model sector.

---

## Editorial Decision

published